## Research Article

perinatal; antenatal; global mental health; health services; mental health

**Corresponding author:**
Raquel Catalao;
Email: raquel.catalao@kcl.ac.uk

# Identification of psychosocial problems in routine antenatal care in Ethiopia: A facility-based cross-sectional study

Raquel Catalao[1] , Charlotte Hanlon[2,3,4], Tigist Eshetu[3], Girmay Medhin[5], Ahmed Abdella[6], Adiyam Mulushoa[4], Tesera Bitew[7], Roxanne C. Keynejad[8], Alemayehu Bekele[3], Negussie Deyessa[9], Atalay Alem[4], Abebaw Fekadu[3], Jane Sandall[10], Louise Howard[8] and Martin Prince[11]

[1]Department of Psychological Medicine, Institute of Psychiatry, Psychology and Neuroscience, King's College London, UK; [2]Division of Psychiatry, Centre for Clinical Brain Sciences, University of Edinburgh, UK; [3]Centre for Innovative Drug Development and Therapeutic Trials for Africa (CDT Africa), College of Health Sciences, Addis Ababa University, Ethiopia; [4]Department of Psychiatry, WHO Collaborating Centre for Mental Health Research and Capacity Building, School of Medicine, College of Health Sciences, Addis Ababa University, Ethiopia; [5]Aklilu-Lemma Institute of Pathobiology, Addis Ababa University, Ethiopia; [6]Department of Obstetrics and Gynaecology, School of Medicine, College of Health Sciences, Addis Ababa University, Ethiopia; [7]Department of Psychology, College of Education and Behavioural Sciences, Injibara University, Ethiopia; [8]King's Women's Mental Health, Institute of Psychiatry, Psychology and Neuroscience, King's College London, UK; [9]Department of Preventive Medicine, School of Public Health, Addis Ababa University, Ethiopia; [10]Department of Women and Children's Health, School of Life and Population Science, King's College London, UK and [11]King's Global Health Institute, King's College London, UK

## Abstract

In this facility-based cross-sectional survey in primary health care centres in southern Ethiopia, women attending for antenatal care (ANC) were screened for depression (PHQ-9; score ≥ 5 and functionally impaired), anxiety symptoms (GAD-7; score ≥ 10), post-traumatic stress disorder (PCL-5 score ≥ 31), intimate partner violence (IPV; non-graphic language screening test) and risky substance use (ASSIST). Clinical notes were reviewed post-consultation for evidence of health worker recognition of psychosocial concerns and actions taken. Of the 2,079 interviewed women, 24.9% had at least one psychosocial problem, and 7.3% had two or more. The most common psychosocial problem was probable IPV (n = 289; 13.9%, 95% CI: [12.57–15.25]), followed by risky khat use (n = 134; 6.5%, 95% CI: [3.72–10.94]) and depression (n = 110; 5.3%, 95% CI: [3.98–7.00]). Identification by ANC professionals was low: 2.7% of women with probable depression, none with probable IPV had it recorded. A history of mental health problems was not documented in 99.7% and only 47.3% (n = 938) reported being asked how they were feeling emotionally. Systemic changes to provider training and procedures are required to promote person-centred maternal care and improve identification of psychosocial problems in ANC in rural Ethiopia.

## Impact Statements

Person-centred maternal care, which respects and responds to women's preferences and values, is essential, but depends on healthcare providers' ability to recognise and respond to psychosocial difficulties. In this facility-based cross-sectional study in antenatal care, the first from Ethiopia and one of the few from a low- and middle-income country to use validated measures in a large sample of women, psychosocial problems (mental health conditions, intimate partner violence and substance abuse) were common, but detection was very low. Fewer than half of pregnant women were asked about their emotional well-being despite finding the enquiry acceptable. Our study highlights the high unmet need for perinatal psychosocial care in low- and middle-income settings and the requirement for systemic changes in health services delivery to promote person-centred maternal care and improve outcomes for all women.

## Introduction

Early identification of psychosocial problems among expectant parents and targeted supportive interventions are crucial for families' well-being (Gram et al., 2024). Person-centred maternal care, defined as care that is "respectful of, and responsive to, women's preferences, needs, and values" (Sudhinaraset et al., 2017), has been highlighted by the World Health Organization (WHO) as a core component of high-quality maternal care (Tunçalp et al., 2015). Person-centred maternal care depends on health workers being able to recognise and

respond to mental health problems and intimate partner violence (IPV) in women attending for antenatal and postnatal care. In low- and middle-income countries (LMICs), perinatal depressive, anxiety and somatic symptoms are estimated to affect 15.6% of pregnant and 19.8% of postnatal women, reflecting a higher burden than in high-income countries (Fisher et al., 2012a). Few studies have examined traumatic stress symptoms in perinatal women, but existing studies from mostly high-income settings indicated a prevalence of 3.3% antenatally and 4.0% postnatally, with higher estimates among women with exposure to childhood maltreatment (Yildiz et al., 2017). IPV is common across cultures and settings, peaking in pregnancy in some countries (WHO, 2005). IPV is itself a potent risk factor for poor maternal mental health (Dadi et al., 2020a). IPV and maternal depression are independently associated with worse obstetric outcomes (Rahman et al., 2004; Da Thi Tran et al., 2022). Maternal depression has been associated with reduced attendance for routine maternal care and increased emergency presentations (Bitew et al., 2016, 2017b), prolonged labour and other perinatal complications (Hanlon et al., 2009; Bitew et al., 2017a), as well as adversities of infant health, growth and development (Dadi et al., 2020b, 2020c). In Ethiopia, maternal depression and IPV act synergistically to increase the risk of infant mortality (Deyessa et al., 2010).

Routine maternal care settings provide ideal opportunities for recognition of maternal psychosocial problems: this is a time when women have most contact with the health system and when intervention can benefit both the woman and her unborn or newborn child. However, most women with perinatal depression in LMICs never receive effective interventions (Prom et al., 2022). The treatment gap for women with high postnatal depressive symptoms in rural Ethiopia was 94% (Azale et al., 2016). In that study, 69.9% of women with postnatal depression thought that they would benefit from intervention, and 49.8% said that maternal care would be the most acceptable place to receive an intervention. In a qualitative study from Ethiopia, midwives, health officers and nurses working in maternal care recognised manifestations of psychosocial problems in perinatal women and felt well-placed to intervene (Bitew et al., 2020). Women with high levels of perinatal depressive symptoms identified barriers to discussing psychosocial problems in routine maternal care settings, including fear of stigma and negative treatment by health workers and perceptions that ANC providers can do little to help (Bayouh, 2020) Little is known about the current levels of identification of and response to psychosocial problems in women attending routine antenatal care in LMICs.

The primary aim of our study was, therefore, to investigate the identification of IPV and depressive, anxiety and traumatic stress symptoms and substance misuse in women attending antenatal care services in a predominantly rural area of southern Ethiopia. As a secondary aim, we investigated factors associated with enquiry about psychosocial concerns by ANC professionals.

## Methods

This study focused on antenatal care in Ethiopia and was part of the wider National Institute for Health and Care Research (NIHR)-funded Global Health Unit on Health System Strengthening in Sub-Saharan Africa (ASSET) project, which aimed to co-develop, test and evaluate health system strengthening interventions to promote integrated, person-centred care across three healthcare platforms

(maternal and newborn care, surgical care and integrated primary care) in four African countries (Ethiopia, Sierra Leone, South Africa and Zimbabwe) (Seward et al., 2022).

### Study design

We conducted a cross-sectional survey and review of clinical documentation of women attending primary care health centres for antenatal care in Central Ethiopia Region (Eshetu et al., 2025).

### Setting

The study was carried out in eight primary health care (PHC) facilities (health centres) in south-central Ethiopia, in Butajira town, Sodo and South Sodo, and West Meskan and Meskan districts, Gurage Zone, Central Ethiopia Regional State (Eshetu et al., 2025).

Most maternal care (antenatal, delivery and postnatal care) for this population is delivered by midwives, health officers and nurses working in PHC health centres and collaborating with community-based health extension workers for outreach (e.g., if women drop out of antenatal care). Coverage of antenatal care has increased substantially in recent years, to an estimated 81.6% (Gebrekirstos et al., 2021). Only high-risk women are referred for ANC at the secondary care level. In the study area, secondary maternal care is available at one primary hospital (Buei) and one general hospital (Butajira), both centrally located in towns.

In the study area, PHC workers, including those staffing the ANC clinics, have been trained and supported to deliver first-line mental health care as part of implementation research projects (Seward et al., 2022) and in line with the Ethiopian National Mental Health Strategy (Federal Democratic Republic of Ethiopia Ministry of Health, 2012). Health centre staff in the study area were also trained using the Ethiopian Primary Healthcare Clinical Guidelines (EPHCG) (Ethiopian Ministry of Health, 2019). The EPHCG is based on the Practical Approach to Care Kit, developed in South Africa and used in Botswana, Nigeria and Brazil (Cornick et al., 2018). The EPHCG was implemented in 400 PHC settings in Ethiopia in 2017, followed by a training cascade, scaling up to all 3,724 health centres in the country during 2019 (Feyissa et al., 2018). The EPHCG is intended to integrate best practice guidelines for all common presentations to PHC facilities in one guide. As part of the integrative approach taken by the EPHCG, guidance for routine antenatal and postnatal care includes assessment of, and interventions for, women's mental health and substance use and psychosocial stressors (including IPV) alongside addressing her reproductive health needs. EPHCG training takes a facility-based team approach based on adult learning principles. Training is structured through case studies that promote familiarisation with the EPHCG guidelines. However, the EPHCG roll-out did not include training on new content areas such as psychosocial care for women during the perinatal period.

### Sampling and recruitment

Recruitment took place from 18 July 2019 to 9 January 2020 at eight health centres, which were selected based on their high ANC patient flow and location, and to obtain a balance of rural and urban districts. Consecutive women attending for routine ANC were approached by research staff, informed about the study and invited to participate. The inclusion criteria comprised all pregnant women

attending any antenatal appointment as part of routine antenatal care, independent of gestation stage, fluency in Amharic (the working language of the study area and region) and aged 18 years or above, who were able to provide informed consent. Women were excluded if they required emergency medical treatment. Women were reimbursed for their time ($2).

## Sample size

In order to measure the identification of 10% of psychosocial problems by PHC workers with a precision of ±5%, with alpha = 0.05 and power of 80%, we needed to recruit at least 139 women with psychosocial problems. Assuming a 20% prevalence of high depressive symptoms and/or exposure to violence in antenatal women (Deyessa et al., 2009; Servili et al., 2010), a total sample of 695 (139 × 0.2) would be required. Assuming an intra-cluster correlation of 0.02 and $n$ = 100 women recruited per facility in eight health centres, the design effect will be $1 + (100 − 1) × 0.02 = 2.98$, and the total target sample was $2.98 × 695 = 2,071$.

## Data collection

Data collection was carried out in a private room within the health facility after the woman's consultation with health workers. Female Lay data collectors, with a minimum of high school education, administered the fully structured questionnaires. Research nurses extracted information from the medical records for each woman using a bespoke template developed for the study. Training was provided to orient data collectors, supervisors and field coordinators to the questionnaires (Eshetu et al., 2025). Supervisors carried out daily data checks at their respective health centres, and field coordinators downloaded and backed up the data every day.

## Measures

### Data from clinical records
The data collectors used a structured form to extract collected clinical data from the women's antenatal care visit. This included parity, gravidity, documented mental health or substance use conditions and their management, documented violence exposure and any interventions.

### Interviewer-administered questionnaires
The following fully structured measures were administered in an interview format.

a) *Sociodemographic characteristics:* including age, educational level, marital status, place of residence and religion.
b) Structured questions about person-centred antenatal care, including whether mental health was asked about and whether this was or would be acceptable. Questions included "Did the health worker ask you about how you were feeling (emotionally) today?" "If yes, how comfortable were you with the health worker asking you about your feelings?" and "If no, would you have liked the health worker to ask about your feelings?." Likert scales were used to collect responses. Women were also asked about their partner's substance use.
c) *Depressive symptoms* were measured using the Patient Health Questionnaire (PHQ-9) (Kroenke and Spitzer, 2002). The PHQ-9 has been culturally validated among primary care attendees (Hanlon et al., 2015) and in pregnant women attending ANC (Girma, 2013) in rural Ethiopia. A cut-off

of five or more has been shown to give optimal sensitivity and specificity against a gold standard of major depressive disorder for primary care attendees, and a cut-off of four or more was optimal for ANC attendees. However, at these cut-offs (in a similar setting), the positive predictive value was low and likely identified women with time-limited, situational distress (Hanlon et al., 2015). For this study, therefore, we considered a woman to have "probable depression" if she scored five or more on the PHQ-9 and reported that she had found it "very difficult" or "extremely difficult" to function in the preceding 2 weeks. Structured questions about help-seeking for depressive symptoms and acceptability of integrated depression care were also administered for those who scored five or more on the PHQ-9.

d) *Anxiety symptoms* were measured using the Generalised Anxiety Disorder (GAD-7) scale. Women were categorised as having low anxiety symptoms if they scored 5–9; moderate symptoms if they scored 10–14 and severe symptoms if they scored 15 or above. We considered a woman to have "moderate-to-severe" anxiety symptoms if the GAD-7 score was 10 or above (Spitzer et al., 2006).

e) *Post-traumatic stress disorder (PTSD)* was measured using the PTSD checklist for DSM-5 (PCL-5) (Blevins et al., 2015), translated into Amharic and adapted for the rural Ethiopia context (Ng et al., 2021). The PCL-5 checklist consists of 20 statements rated from "not at all" (score 0) to "extremely" (score 4). Each item rated 2 ("moderately") or higher was considered endorsement of a symptom. PCL-5 divides PTSD symptoms, with questions 1–5 reflecting criterion B (intrusion), questions 6–7 reflecting criterion C (avoidance), questions 8–14 reflecting criterion D (negative alterations in cognitions and mood) and questions 15–20 reflecting criterion E (alterations in arousal and reactivity). Women met DSM-5 criteria for "probable PTSD" if they scored 31 or above on PCL-5 and endorsed at least one criterion B, C, D or E symptom.

f) *Intimate partner violence (IPV)* was measured using the nongraphic language Intimate Partner Violence Screening test (Zink et al., 2007) based on the Revised Conflict Tactics Scales (Straus et al., 1996). This measure contains five questions on engagement in, or experience of, physical or psychological violence with an intimate partner. This scale was previously used in the study population and found to be acceptable and to have convergent validity with the more extensive WHO IPV questionnaire (Bitew et al., 2016). A score of two or more on questions 1 (working out arguments), 3 (partner treatment) or 4 (feeling safe) indicated marital discord and "probable IPV."

g) *Substance use*, namely alcohol and khat consumption, was measured using the World Health Organisation ASSIST questionnaire (WHO, 2010). The ASSIST has been used previously in the Ethiopian setting following rigorous translation procedures (Ambaw et al., 2017). The ASSIST allows classification of the level of risk associated with each substance reported to be used (low, medium and high) based on the total score. Khat is an amphetamine-like plant material that can cause dependence. Chewing the leaves of the khat shrub is common in certain countries of the Arabian Peninsula and East Africa, including Ethiopia (Ayano et al., 2024).

h) *Social support* was assessed using the Oslo Social Support scale (OSS-3), which was used previously in the study setting and found to have convergent validity (Fekadu et al., 2014).

### Data analysis

Data were analysed using Stata software version 14 (Statacorp, 2017). Descriptive statistics were used to summarise frequencies for categorical variables. Prevalence rates and confidence intervals, adjusted for health centre clustering, were calculated. Cross-tabulations were used to identify comorbidity between psychosocial conditions, and chi-squared tests were performed to test the significance of associations. Poisson regression working models were used to calculate prevalence ratios between enquiries about women's emotional well-being and demographic and clinical characteristics. To account for the effect of the number of data collection days in each health centre and average ANC attendance per health centre per day (calculated over a 6-month period), we included these variables and used health centre as the clustering variable in our regression models. Complete case analyses were performed.

### Results

Out of 2,426 women attending for antenatal care, 226 women were ineligible. Reasons were inability to converse in Amharic (*n* = 203), being younger than 18 years (*n* = 14) or requiring emergency treatment (*n* = 9). Out of 2,200 eligible women, 2,079 (94.5%) consented to participate. Of non-participating women, 80 (3.6%) declined, and 41 (1.9%) did not attend their scheduled assessment.

### Characteristics of the sample

Table 1 summarises the sample's socio-demographic characteristics. Over one-third of women (36.1%) were younger than 25 years of age. Participants had low levels of education; 29.4% had no formal education and 53.8% had only primary education. Almost all participants (94.8%) were married, but only one quarter (25.2%) reported strong social support. Nearly a quarter (24%) of women reported that their husbands chewed khat daily or at a level that they considered to be too high. Most women (98.3%) were in the second or third trimester of pregnancy, 28.7% were nulliparous and almost all reported that the pregnancy was wanted, either at the outset or after a time.

### Prevalence and comorbidity of psychosocial problems

Screening measures identified psychosocial problems among almost one quarter of women (*n* = 517, 24.9%). The most common psychosocial problem was probable IPV (13.9%; 95% confidence interval (CI): [12.57–15.35]), followed by risky khat use (6.5%; 95% CI: [3.72–10.94]), probable depression (5.3%; 95% CI: [3.98–7.00]), probable PTSD (5.1%; 95% CI: [3.07–8.36]), moderate to severe anxiety symptoms (3.7%; 95% CI: [2.36–5.62]), and risky alcohol use (0.5%; 95% CI: [0.29–0.98]) (Table 2).

Psychosocial problems commonly co-occur with each other (see Table 3 and Supplementary Material S1); 29.2% of women identified with a psychosocial problem had two or more psychosocial problems (7.3% of the total sample); 29.1% of those with probable depression had comorbid moderate-to-severe anxiety symptoms, 22.7% also had probable PTSD and 32.7% had experienced probable IPV. Of the 13.9% of women who screened positive for probable PTSD, 44.3% experienced moderate-to-severe anxiety symptoms and 41.5% had probable IPV.

### Recognition of psychosocial problems

Of those women with probable depression (*n* = 110; 5.3% of total sample), health professionals only documented a psychosocial problem in 2.7% (*n* = 3; see Table 2). Of the 76 women (5.3%) with moderate or severe levels of anxiety symptoms, only three women (3.9%) had documentation of an emotional problem in their health

**Table 1.** Characteristics of participants (*N* = 2,079)

| Characteristic | Categories | *n* (%) |
|---|---|---|
| **Age (years)** | 18–24 | 751 (36.1) |
| | 25–34 | 1,144 (55.0) |
| | 35–50 | 184 (8.9) |
| **Educational level** | No formal education | 611 (29.4) |
| | Primary education | 1,119 (53.8) |
| | Secondary education | 290 (14.0) |
| | Post-secondary | 59 (2.8) |
| **Residence** | Rural | 1,115 (53.6) |
| | Urban | 964(46.4) |
| **Marital status** | Married | 2,013 (96.8) |
| | Single | 37 (1.8) |
| | Separated, divorced or widowed | 29 (1.4) |
| **Religious affiliation** | Muslim | 1,131 (54.4) |
| | Orthodox Christian | 771 (37.1) |
| | Protestant Christian | 177 (8.5) |
| **Parity (*n* = 2,076)** | 0 | 595 (28.7) |
| | 1 | 485 (23.4) |
| | 2–4 | 777 (37.4) |
| | 5 or more | 219 (10.6) |
| **Gestation (*n* = 2,062)** | 1st trimester | 34 (1.7) |
| | 2nd trimester | 1,120 (54.3) |
| | 3rd trimester | 908 (44.0) |
| **Whether pregnancy was wanted** | Wanted | 1,632 (78.5) |
| | Initially unwanted, now wanted | 409 (19.7) |
| | Unwanted | 38 (1.8) |
| **Social support** | Poor | 533 (25.6) |
| | Intermediate | 1,023 (49.2) |
| | Strong | 523 (25.2) |
| **Husband's alcohol use** | Daily/perceived as too much | 155 (7.5) |
| **Husband's khat use** | Daily/perceived as too much | 498 (24.0) |
| **Health centre attended** | A | 852 (41.0) |
| | B | 285 (13.7) |
| | C | 379 (18.2) |
| | D | 104 (5.0) |
| | E | 138 (6.7) |
| | F | 158 (7.6) |
| | G | 48 (2.3) |
| | H | 115 (5.5) |

**Table 2.** Detection of psychosocial problems by ANC providers (n = 2,079)

| Psychosocial problem | Prevalence n (%) | 95% Confidence interval adjusted for health centre clustering | Identified and documented in clinical records n, % |
|---|---|---|---|
| **Depressive symptoms** | | | |
| PHQ–9 total score | | | |
| 0–4 | 1,284 (61.8) | | 0 (0.0) |
| 5–9 | 690 (33.2) | | 1 (0.1) |
| 10–14 | 80 (3.9) | | 2 (2.5) |
| 15 and above | 25 (1.2) | | 0 (0.0) |
| Probable depression (PHQ–9 ≥ 5 and very/extremely high functional impairment) | 110 (5.3) | (3.98–7.00) | 3 (2.7) |
| **Anxiety symptoms** | | | |
| GAD–7 total score | | | |
| 0–4 | 1752 (84.3) | | 0 (0.0) |
| 5–9 | 251 (12.0) | | 1 (0.4) |
| 10–14 | 56 (2.7) | | 2 (3.6) |
| 15 and above | 20 (1.0) | | 0 (0.0) |
| GAD–7 ≥ 10 | 76 (3.7) | (2.36–5.62) | 3 (3.9) |
| **Post-traumatic stress symptoms** | | | |
| PCL–5 ≥ 31 | 89 (4.3) | | 2 (2.3) |
| PCL–5 Probable PTSD (DSM criteria) | 106 (5.1) | (3.07–8.36) | 1 (0.9) |
| **Probable intimate partner violence** | | | |
| IPV scale - >1 on items 1, 3 or 4 | 289 (13.9) | (12.57–15.35) | 0 |
| **Substance use problem** | | | |
| ASSIST moderate/high risk of khat use | 134 (6.5) | (3.72–10.94) | 0 |
| ASSIST moderate/high risk of alcohol use | 11 (0.5) | (0.29–0.98) | 0 |

**Table 3.** Co-occurrence of psychosocial problems

| N = 2079 | Probable depression | Moderate-to-severe anxiety symptoms | Probable PTSD | Intimate partner violence (IPV) | Risky khat use |
|---|---|---|---|---|---|
| **Probable depression (n = 110)** | | 32 (29.1%)*** | 25 (22.7%)*** | 36 (32.7%)*** | 7 (6.4%) |
| **Moderate-to-severe anxiety symptoms (n = 76)** | 32 (42.1%)*** | | 47 (61.8%)*** | 32 (42.1%)*** | 10 (13.2%)* |
| **Probable PTSD (n = 106)** | 25 (23.6%)*** | 47 (44.3%)*** | | 44 (41.5%)*** | 13 (12.3%)* |
| **Probable IPV (n = 289)** | 36 (12.5%)*** | 32 (11.1%)*** | 44 (15.2%)*** | | 33 (11.4%)*** |
| **Risky khat use (n = 134)** | 7 (5.2%)*** | 10 (7.5%)* | 13 (9.7%)* | 33 (24.6%)*** | |

*Note*: PTSD = post-traumatic stress disorder; chi-square test, ***$P < 0.001$; *$P < 0.05$.

records. None of the women identified *via* screening questionnaires had documentation of any concern about probable IPV, risky khat or alcohol use. There was no documentation of a history of mental health or substance use problems in 99.7% and 98.3% of the sample, respectively (Supplementary Material S2).

Fewer than half of participants (n = 983; 47.3%) reported being asked how they were feeling emotionally by their ANC provider. The vast majority of those asked (n = 932; 98.4%) reported feeling comfortable with the enquiry. Of those not asked (n = 1,096), 58.4% reported they would have liked to be asked how they were feeling emotionally (see Supplementary Material S2). In women with depressive symptoms (n = 795), the majority (n = 741;93.2%) were not asked about those symptoms, despite 56.0% (n = 445) reporting it was highly acceptable for health workers to ask (Table 4).

### Factors associated with psychosocial enquiries

For those 983 women (47.3% of the total sample) asked how they were feeling emotionally at ANC, we investigated factors associated with enquiry about psychosocial concerns by ANC professionals. Women living in an urban environment (prevalence ratio (PR) = 0.71; 95% CI: [0.65–0.78]) and those with no previous children (PR = 0.88; 95% CI: [0.79–0.99]) were less likely to be asked how they were feeling emotionally compared to those living in rural areas and those with children (see Table 5). Women in the third trimester of pregnancy were more likely to be asked how they were feeling emotionally than those in the second trimester (PR = 1.37; 95% CI: [1.25–1.50]), and those older, in the age range 35–50 years were also more likely to be asked (PR = 1.18; 95% CI: [1.02–1.36]) compared to those aged 25–34 years. There were no

**Table 4.** Care for women with depressive symptoms (PHQ > =5; *n* = 795)

| Aspect of care | Categories | *n* (%) |
|---|---|---|
| **Asked about depressive symptoms** | Yes | 54 (6.8) |
| | No | 741 (93.2) |
| **Acceptability of being asked about depressive symptoms by a health worker** | Not at all acceptable | 62 (7.8) |
| | Not very | 47 (5.9) |
| | Neutral | 51 (6.4) |
| | Somewhat | 190 (23.9) |
| | Highly acceptable | 445 (56.0) |
| **Timing of the onset of depressive symptoms** | Before pregnancy | 72 (9.2) |
| | Since becoming pregnant | 723 (90.9) |
| **Type of help-seeking for depressive symptoms (*n* = 209)** | Traditional healer | 7 (0.9) |
| | Holy water | 15 (1.9) |
| | Religious leader/advisor | 25 (3.1) |
| | Health post | 9 (1.0) |
| | Health centre | 140 (17.6) |
| | Hospital | 27 (3.4) |
| | Pharmacy | 2 (0.3) |
| | Mental health professional | 4 (0.5) |
| | Private clinic | 43 (5.4) |
| **Level of satisfaction with care with help obtained (*n* = 208)** | Not satisfied at all | 46 (22.1) |
| | Somewhat satisfied | 90 (43.3) |
| | Satisfied a lot | 72 (34.6) |

Most women reported onset of depressive symptoms since becoming pregnant (90.9%; *n* = 737) and did not seek help for these (94.4%; *n* = 790). For those seeking help, the health centre was the most common place to seek help (17.6%), with a minority seeking help from traditional/religious sources. Most of those seeking help (*n* = 209) were somewhat or very satisfied with help provided (Table 4).

significant differences in enquiry for women with *versus* those without depressive, anxiety or PTSD symptoms, probable IPV or substance misuse.

## Discussion

In this facility-based cross-sectional study of women attending primary care health centres for ANC in south central Ethiopia, psychosocial problems (including mental health conditions, IPV and risky substance use) were common: 24.9% screened positive for at least one problem, and there were high levels of co-occurrence. However, identification and documentation of psychosocial problems by ANC providers was low, with fewer than 5% of women with a psychosocial problem identified. Less than half of the participants were asked about their emotional well-being, despite the majority finding this acceptable.

To improve maternal outcomes, women's psychological and social needs should be explored alongside their obstetric health in efforts to improve integration, quality of care and the delivery of person-centred maternal care (Sudhinaraset et al., 2020). Our study adds to growing evidence of a high unmet need for perinatal psychosocial care in LMICs, where maternal depression, anxiety (Fisher et al., 2012b) and IPV (Ma et al., 2023) exposure are more prevalent than in high-income countries.

**Table 5.** Factors associated with women being asked how they were feeling emotionally at the ANC appointment

| Total = 983 (47.3% of total sample) | Women asked how they were feeling emotionally (*n*, %) | Poisson regression models Prevalence ratio (95% confidence interval) adjusted for health centre clustering |
|---|---|---|
| **Age (years)** | | |
| 18–24 | 349 (46.5%) | 1.00 (0.90–1.10) |
| 25–34 | 533 (46.6%) | (ref) |
| 35–50 | 101 (54.9%) | 1.18 (1.02–1.36)* |
| **Education level** | | |
| No formal education | 319 (52.2%) | (ref) |
| Primary education | 492 (44.0%) | 0.84 (0.76–0.93)* |
| Secondary education | 143 (49.3%) | 0.94 (0.82–1.08) |
| Post-secondary | 29 (49.1%) | 1.08 (0.83–1.42) |
| **Residence** | | |
| Rural | 608 (54.5%) | (ref) |
| Urban | 375 (38.9%) | 0.71 (0.65–0.78)** |
| **Parity (*n* = 2,076)** | | |
| 0 | 261 (43.9%) | 0.88 (0.79–0.99)* |
| 1 | 215 (44.3%) | 0.89 (0.79–1.00) |
| 2–4 | 388 (49.9%) | (ref) |
| 5 or more | 118 (53.9%) | 1.08 (0.94–1.24) |
| **Gestation (*n* = 2,062)** | | |
| 1st trimester | 18 (52.9%) | 1.31 (0.95–1.81) |
| 2nd trimester | 455 (40.6%) | (ref) |
| 3rd trimester | 502 (55.3%) | 1.37 (1.25–1.50)** |
| **Psychosocial problems** | | |
| Probable depression (*n* = 110) | 59 (53.6%) | 0.91 (0.83–1.00) |
| Moderate-to-severe anxiety symptoms (*n* = 76) | 158 (48.3%) | 1.00 (0.79–1.28) |
| Probable PTSD (*n* = 106) | 47 (44.3%) | 0.93 (0.75–1.16) |
| Probable IPV (*n* = 289) | 121 (41.9%) | 0.87 (0.75–1.00) |
| Risky khat use (*n* = 134) | 52 (38.8%) | 0.81 (0.65–1.00) |

*\*P < 0.05; \*\*P < 0.001.*

Although probable IPV was the most prevalent psychosocial problem in our study, the prevalence estimate (13.9%) is lower than in other antenatal studies from Ethiopia, where IPV prevalence was reported to be 21.0% (Belay et al., 2019) and 26.1% (Alebel et al., 2018). In a previous study in the same region of the country, 77% of pregnant women reported experiencing physical violence (Gossaye et al., 2004). We found high co-occurrence of IPV and mental health conditions in keeping with previous studies in Ethiopia (Belay et al., 2019). IPV strongly predicts persistence of antenatal depressive symptoms and incidence of postnatal depressive symptoms in this study setting (Bitew et al., 2019). The consequences of unrecognised and untreated mental

health conditions and psychosocial risks, especially IPV, are substantial (Gebreslasie et al., 2024). In our study, no women experiencing probable IPV were identified by their ANC provider, highlighting the scale of missed opportunities to identify IPV and intervene.

Similarly, only three of the 110 women identified as experiencing probable depression were identified by health professionals, highlighting a large detection gap. This is despite extensive evidence for adverse obstetric and neonatal outcomes associated with antenatal depression in Ethiopia. These include low birth weight in some (Beyene et al., 2021) but not all studies(Hanlon et al., 2009), delayed initiation of breastfeeding, increased risk of infant diarrhoea and child accidental injuries (Ross et al., 2011). In this region of Ethiopia, maternal depression, when present in addition to intimate partner violence, was associated with increased child mortality (Deyessa et al., 2010). Whereas much has been written on the large treatment gap for perinatal depression, estimated to be over 90% in Ethiopia (Azale et al., 2016), our study suggests that low recognition and action by ANC providers is an important opportunity for intervention.

Evidence from LMICs indicates that person-centred maternal care improves pregnancy outcomes and increases the woman's willingness to return to the facility for her next delivery (Sudhinaraset et al., 2020), whereas its absence is associated with birth and postpartum complications (Raj et al., 2017). Our results highlight that despite the rollout of the EPHCG (adapted PACK) programme in Ethiopia, which comprises the provision of carefully designed, comprehensive and integrated clinical decision support into primary health care (Feyissa et al., 2018), further extensive systemic changes are required to equip health workers with the necessary tools to facilitate identification of psychosocial problems. These include training, support and supervision for health professionals, manageable caseloads and conducive clinical environments. Research from South Africa emphasised that efforts to improve recognition of mental health conditions and IPV must be paired with pathways for referral and treatment or support (Abrahams et al., 2023). In the Ethiopia ASSET study, further facility-based training on person-centred maternal care has been developed in collaboration with the Federal Ministry of Health of Ethiopia and evaluation analyses are underway (Eshetu et al., 2025). Ongoing research in Ethiopia is also developing brief psychological interventions for perinatal women experiencing IPV, which can be feasibly delivered by primary care staff (Keynejad et al., 2020, 2024).

Qualitative work with health professionals is essential to explore further barriers to detecting and documenting psychosocial problems. This is likely to be a multi-factorial problem, including lack of confidence in enquiring, lack of referral pathways if disclosure occurs, but also time constraints to accurate documentation. There is evidence of a high burden of documentation on health workers, and the Ethiopian government has designed initiatives to improve Health Information Services (Biru et al., 2022; Tilahun et al., 2025). Future government initiatives to expand the use of digital tools could facilitate these processes, creating opportunities to incorporate person-centred care into informatics innovations.

Further collaborative, co-produced research is required to understand how services can empower women with a greater understanding of their rights and instil trust to facilitate disclosure of psychosocial problems. Local qualitative research found that women valued respectful and responsive communication and this affected their willingness to disclose psychosocial problems (Eshetu et al., 2025).

## Strengths and limitations

This is the first study to compare clinical identification of psychosocial problems in antenatal care in Ethiopia with detection using several locally validated measures in a large, representative sample. Limitations include the potential for under-disclosure of IPV and mental health symptoms due to stigma, as well as under-detection of anxiety and trauma symptoms due to the use of scales not extensively validated in this setting. We also lacked data on the ethnicity of the women. It is possible that ANC providers detect psychosocial problems but do not routinely document them in clinical records, highlighting the need for interventions to improve the quality of data recording alongside clinician training and referral pathways for treatment and support.

## Conclusions

Despite the high prevalence of psychosocial problems and their co-occurrence in antenatal care in rural Ethiopia, recognition by health professionals is very low. Fewer than half of pregnant women were asked about their emotional well-being despite finding the enquiry acceptable. Further research is required to understand what systemic changes can promote person-centred maternal care and improve recognition, documentation and response to psychosocial problems, to improve outcomes for all women.

**Open peer review.** To view the open peer review materials for this article, please visit http://doi.org/10.1017/gmh.2026.10221.

**Supplementary material.** The supplementary material for this article can be found at http://doi.org/10.1017/gmh.2026.10221.

**Data availability statement.** The data that support the findings of this study are openly available in The Open Science Framework at https://osf.io/a8hbg.

**Acknowledgements.** The authors would like to thank all the women who kindly agreed to participate in this study.

**Author contribution.** RC: Data curation, methodology, formal analysis, writing – original draft preparation. CH: Conceptualisation, supervision, methodology, funding acquisition, writing – reviewing and editing. TE: Investigation, project administration, resources, writing – reviewing and editing. GM: Methodology. AA: Project administration, investigation, writing – reviewing and editing. AM: Investigation, writing – review and editing. TS: Writing – reviewing and editing. RK: Writing – reviewing and editing. AB: Investigation, writing – reviewing and editing. ND: Writing – reviewing and editing. AA: Writing – reviewing and editing. AF: Writing – reviewing and editing. JS: Writing – reviewing and editing. LH: Writing – reviewing and editing. MP: Methodology, writing – reviewing and editing, supervision.

**Financial support.** The research underpinning the findings presented in this article was funded by the National Institute of Health and Care Research (NIHR) Global Health Research Unit on Health System Strengthening in Sub-Saharan Africa (ASSET), King's College London (GHRU 16/136/54) using UK aid from the UK Government. CH and MP are funded by an NIHR global health research group on homelessness and mental health in Africa (HOPE; NIHR134325). The views expressed in this publication are those of the authors and not necessarily those of the NHS, the National Institute for Health and Care Research or the Department of Health and Social Care, England. CH is also funded by the Wellcome Trust through grants 222154/Z20/Z (SCOPE) and 223615/Z/21/Z (PROMISE).

**Competing interests.** The authors declare none.

**Ethics statement.** Ethical approval was obtained from the Institutional Review Board of the College of Health Sciences, Addis Ababa University (Reference number: 028/18/Psy) and King's College London (Reference

number: HR-17/18–6,570). Any woman who was identified as having severe depression or who endorsed the item indicating suicidal ideation on PHQ-9 was flagged to the antenatal care provider to allow for consideration of clinical intervention. For women who disclosed exposure to IPV during pregnancy, research staff were trained to listen non-judgementally, offer privacy and confidentiality and information about local agencies that assist in keeping with the World Health Organization guidelines (WHO, 2005).

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
