## [Reviewer Report]

Quite well designed and conducted study in a LMIC. Important piece of work. Well written. However, I have two minor points i wish to see as edits that will complete the paper. Congratulations the whole team.

1. Please describe what is Khat. Even i did not know it, and googled. I believe this is it. Khat (Catha edulis) is a flowering evergreen shrub native to East Africa and the Arabian Peninsula, whose leaves are chewed for their stimulant, amphetamine-like effects. It is considered a dangerous substance with high potential for abuse, often leading to significant physical, mental, and social health consequences.

2. “Therefore, the target sample was 2071”. I could not work out how this figure was calculated. There is reference to 139, then 100 and 7 centres but how did 2071 came there. Please make it more clear

---

## [Reviewer Report]

It is quite difficult to follow how the sample has been subsetted to answer the study questions. Some parts are difficult to follow up when there is no exact reference of N total for each subset sample, particularly in Table 5 (page 18).

Check the numbers and percentages, particularly for Parity and Gestation; the sum does not match N provided in the table.

For clarity, it might be helpful to add a flowchart explaining the subsetting process.

In Table 4, two categories (type of help-seeking and level of satisfaction) do not sum up the total sample indicated in the table name.

In Table 2, the category “Probable depression (PHQ-9 ≥5 and very/extremely high functional impairment)" is not clear when given the description explained in the PHQ-9 score category.

After women were identified with anxiety, depression, and PTSD symptoms, did they receive a referral or follow the care pathway in the health center? Women identified with suicidal ideation also were part of the study?

Table 1.

May you add the “Urban” subcategory to have the complete table?

Were pregnant women asked about their cultural group of precedence? Because sometimes minority groups could have major difficulties accessing healthcare.

Maybe, since there is a clustering by health center, the frequency of participants per center can be explained in the characteristics table.

The tests GAD-7 and Partner Violence Screening test were culturally adapted and validated for the setting. If not, how did the team adapt or ensure the appropriate use and application of the instruments?

On page 12, (b), how were these questions collected? Maybe using a questionnaire with a Likert scale or a different approach was taken?